# Design Framework of a Traceability System for the Rice Agroindustry Supply Chain in West Java

**Pradeka Brilyan Purwandoko [1]** **, Kudang Boro Seminar [1,\*]** , Sutrisno [1] and Sugiyanta [2]

[1] Department of Mechanical and Biosystem Engineering, Faculty of Agricultural Technology, IPB University, Bogor 16680, West Java, Indonesia; pradekabrilyan@gmail.com (P.B.P.); kensutrisno@yahoo.com (S.)

[2] Department of Agronomy and Horticulture, Faculty of Agriculture, IPB University, Bogor 16680, West Java, Indonesia; mr_sugiyanta@yahoo.co.id

\* Correspondence: seminarkudangboro@gmail.com; Tel.: +62-8164834625

**Abstract:** Rice is a vital food commodity in Indonesia due to its role as a staple food for most Indonesian people. The rice supply chain in Indonesia varies from one region to another and it is difficult to trace movement along the chain from land to customers. This introduces non-transparency and uncertainty in the quantity and quality of rice at every node along the supply chain. The crucial issues of food safety and security, as well as consumer concern and curiosity in buying and consuming foods, increases the need for a traceability system for the rice value chain which can be easily and widely accessed. This paper describes the design framework of an IT (Information Technology)-based traceability system for the rice supply chain on web platforms. The system approach has been followed (where the system requirements are identified based on supply chain characteristics) and then the logical framework for implementing internal and external traceability was modeled using IDEF-0 (Integrated Definition Modeling). This paper further presents an explanation of the ERD (Entity Relationship Diagram) as an initial step to modeling the data requirements and a model of information exchange between stakeholders that explains the data that must be recorded and forwarded to the next stakeholder. Finally, we propose the CBIS (Computer Based Information System) concept to develop a traceability system in the rice supply chain.

**Keywords:** design framework; rice; supply chain; traceability system

## 1. Introduction

Rice is widely planted and is the staple food of around 3.5 billion people worldwide, and Indonesia is one of the countries with the highest consumption rates in the world, reaching 38.41 million tons in 2012 [1,2]. In Indonesia, rice accounts for 45% of the total nutritional intake needed or about 80% of the primary carbohydrate source in the consumption patterns of the Indonesian people [3]. Therefore, aspects of quality and food safety are essential things that must be considered. However, in reality, the rice production chain in Indonesia still faces several obstacles, and one of them is the quality manipulation carried out by specific stakeholders. Several previous studies conducted by Suismono and Ramli et al. [4,5] found rice that was sprayed with aromatic compounds and bleached with uncontrolled concentrations where it could harm the health of consumers. Also, cases of counterfeiting of rice occur where traders or rice mills mix rice between varieties and qualities.

Another important issue in the rice supply chain is the unavailability of data that documents all activities, starting from production and continuing with processing and distribution. One of the suggested concepts to overcome these problems is recording all information about a food product in a traceability system. Based on European Union regulations (178/2002), traceability is defined as the ability to track and follow any food, feed, food producing animal, and other substances that

will be used for consumption in all stages of production, processing and distribution [6–8]. In its implementation, traceability systems in several countries including Indonesia have not been carried out thoroughly and are still under the authority of each stakeholder, so cases of quality manipulation, production errors, and other cases will be difficult to detect [9,10]. Furthermore, this problem will be detrimental to business actors (stakeholders) because consumers are hesitant to buy food products when there is no detailed information regarding the quality and safety of these products [10]. It is crucial to develop Information Technology (IT)-based traceability systems to overcome this problem in the rice supply chain.

In the context of information exchange, IT-based traceability systems would support the communication of important information and integrate rice supply chains that are globally dispersed [11]. Nevertheless, the absence of a standard format for recording the information is the biggest challenge to realizing information exchange and integration between stakeholders along the supply chain. The design framework provides guidelines to stakeholders in the supply chain to streamline their operation processes with each other to implement and maintain traceability [11]. This study aimed to develop a design framework to implement internal and external traceability in the rice supply chain. Firstly, the usage requirements of the system were defined using use-case diagrams and IDEF-0 (Integrated Definition Modeling) to show its implementation at each stakeholder. Furthermore, modeling the data requirements is described as the first step in creating the database, and a sequence diagram was used to model the information exchange between stakeholders. Finally, the CBIS (Computer Based Information System) concept was proposed to support traceability along the rice supply chain.

## 2. Literature Review

ISO 22005 explains that food security is the responsibility of all stakeholders involved in the production process. Therefore, all stakeholders in the supply chain must have the ability to identify who is the supplier (one step backward) and to whom the product is distributed (one step forward) [12]. The development of an IT-based traceability system can be used because it has the ability to follow the historical route of food from upstream to downstream. This system is capable of capturing, storing, and transmitting information about the origin of raw materials, processing, and also all activities carried out by stakeholders in the supply chain so that it can ensure all production practices are carried out under the established standard operating procedures. The use of IT in traceability systems has several advantages, namely: (a) integrating data and information from various stakeholders (multi-users), (b) increasing accuracy in data input, (c) providing the ability to communicate and exchange information between stakeholders, and (d) simplifing and speeding up control and monitoring processes [13,14]. Therefore, the use of IT can realize transparency in the supply chain because it is able to systematically manage information related to the products [11].

Nowadays, information exchange has become the main requirement to actualize effective and sustainable supply chains [15]. However, Lam and Postle [16] identified that the standard framework that describes the information exchange process on the supply chain traceability system is limited. In the food supply chain, studies on the development of traceability frameworks have been carried out by several researchers. Zhang et al. [17] developed a traceability system design framework in the tilapia supply chain in China. In this study, stakeholders who play a role in the business process were identified and the supply chain structure was modeled. Furthermore, the information flow and the data that must be recorded by each stakeholder in the supply chain is described and then modeled with the UML (Unified Modeling Language) class diagram. Hu et al. [12] designed a traceability system framework for vegetable supply chains in China. This study was able to identify the structure of the vegetable supply chain and the relationship between stakeholders and traceability systems was modeled using use-case diagrams. A critical point analysis of valuable information was also carried out and then modeled using UML static diagrams. The results also explained the design and architecture of traceability systems in the vegetable supply chain. A system evaluation was also

carried out by Hu et al. [12] for system improvement. From the perspective of the development of a traceability system, it is crucial to overcome the traceability from data and information management, which relies heavily on the framework that has been prepared [11]. However, not all traceability system frameworks are available, primarily in the rice supply chain.

## 3. Methodology

### 3.1. Field Survey

This research was conducted over two years from October 2016 to December 2018 in West Java Province, which is the center of rice production in Indonesia. Firstly, a field survey was conducted on the organic rice agroindustry companies, namely Sarinah Agro Mandiri, Ltd. and Alam Subur, Ltd. Secondly, a field survey was conducted in the anorganic rice agroindustry companies, namely Jatisari Sri Rejeki, Ltd. and Sindang Asih, Ltd. The field survey was conducted to: (1) identify stakeholders involved in the rice supply chain, (2) observe each stage of the production process, and (3) map the needed information that must be recorded at each stakeholder in the supply chain.

### 3.2. Data Collection

The process of data collection in this study was conducted by semi-structured interviews with all stakeholders who play a role in the rice supply chain in West Java Province. Interviews were conducted to find out the business processes in the supply chain and to identify the availability of documentation systems by each stakeholder.

### 3.3. System Analysis

In this study, the Unified Modeling Language (UML) is applied as a system modeling tool. The IDEF-0 model is used to design the framework a of traceability system in the rice supply chain. Furthermore, this paper adopted a use-case diagram, sequence diagram, and Entity Relationship Diagram (ERD) to model traceability system requirements.

## 4. Rice Supply Chain

Supply chains are defined as a system involving various organizations, individuals, technology, multiple activities and resources to channel products in the form of goods or services from suppliers to consumers [18]. In simple terms, a supply chain describes the activities carried out to process the raw materials and additional materials needed to produce added value for the product and to be distributed to customers at a certain price. Chopra and Meindl [19] explained that the supply chain structure involves all stakeholders who play both direct and indirect roles to meet customer needs. West Java Province is one of the centers of rice production in Indonesia, which produced 11 million tons of dried rice grain that contributed around 22% of national rice production in 2014 [20]. This research carried out a study of various stakeholder activities in the supply chain to obtain information on the rice supply chain mechanism in West Java Province, Indonesia. The general rice supply chain mechanism in West Java Province is presented in Figure 1.

Van der Vorst [21] explains that the supply chain structure describes the limits of supply chain networks and describes the principal members and their roles. Based on Figure 1, the structure of the rice supply chain in West Java Province consists of six stakeholders, starting from farmers who carry out rice cultivation activities and sell their crops to middleman or the rice milling industry. Middlemen have the role of collecting grain from farmers and selling it to the industry with defined quality standards based on physical attributes. Several middlemen have small rice milling machines, so that they can sell their rice to retailers (traditional markets). In the processing industry, the harvested dry grain is processed into white rice through the drying and grinding process. The processing industry then sells the products through distributors or directly to retailers. Some of the rice milling industries are BULOG (The Indonesia Logistics Bureau) working partners who are state-owned general

companies engaged in food logistics. This partnership enables the rice milling industry to get an order to procure a certain quota of rice that BULOG will buy at an agreed price. Rice owned by the BULOG is then used as a national rice stock that will be distributed to society and partly sold commercially to retailers.

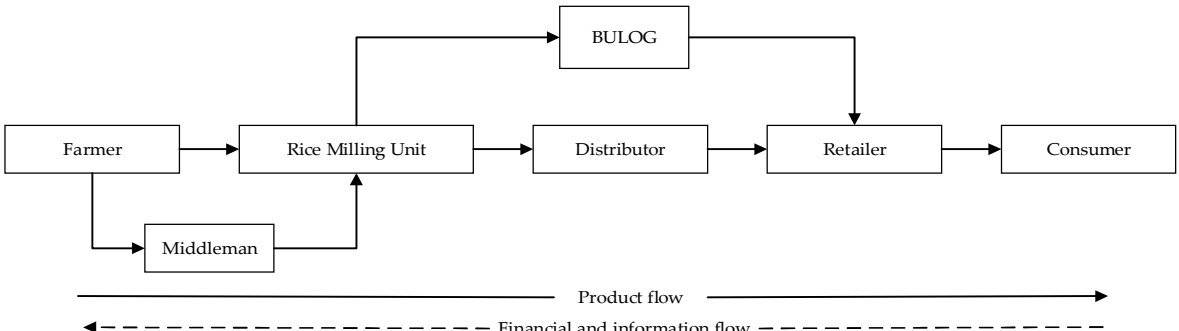

**Figure 1.** Rice supply chain in West Java Province.

## 5. Usage Requirement

The traceability system has the main purpose of recording and documenting food products and materials used during the production process. According to Folinas et al. [22], the traceability system should be integrated into the supply chain and must be able to document and communicate information on product quality, the origin of raw materials, and food safety requirements. Therefore, to develop an efficient rice traceability system, the user requirements in the rice supply chain in the field need to be defined by using a systems approach. The requirements for the use of traceability systems in the rice supply chain are built using Unified Modeling Language (UML) use-case diagrams. Figure 2 shows the use-case diagrams for the development of traceability information systems in the rice supply chain.

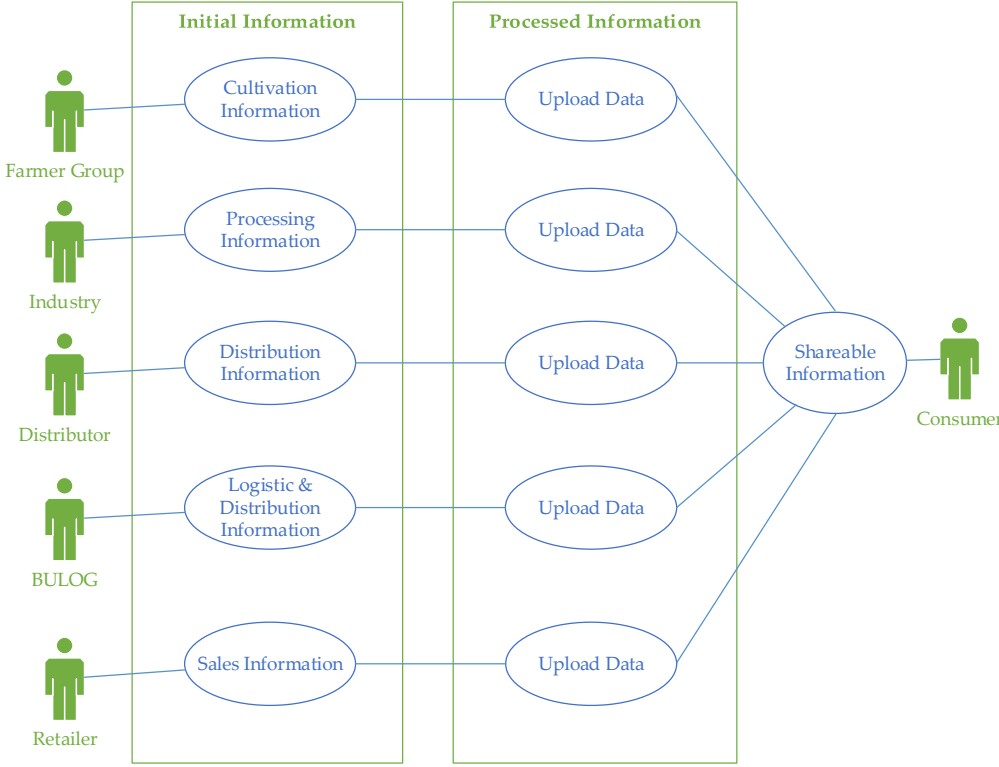

**Figure 2.** Use-case diagram of a rice traceability system.

The use-case diagram describes the behavior of the system. According to Satzinger et al. [23], use-case diagrams describe interactions and functions between one or more actors within the system to be created. Use-case diagrams consist of several components, namely: (1) actors who are objects that interact with the system, (2) a use-case describing the functionality provided by the system as units that exchange messages between systems with actors, (3) relationships that describe interactions that occur between actors and use-cases; and (4) boundary systems that describe system boundaries [24,25]. Based on Figure 2, there is some information recorded by each stakeholder. In necessary information, members of the farmer group record all activities during cultivation and post-harvest such as the variety used, date of planting, fertilization, pest and disease control, harvest mechanisms, and others. The industry must be able to enter details of processing such as procurement data of raw materials, production batches, quality testing, nutritional product content, and others. Distributors manage information about the product distribution process, whereas BULOG manages information about product storage and distribution. Retailers must be able to record information about product sales to consumers. The information entered is then stored in the database system so that the users (consumers) can retrieve product information based on the data available in the system. In the proposed system, we do not include middlemen because they are a type of illegal stakeholder who are not registered in the Ministry of Agriculture or the Ministry of Industry of the Indonesian Republic.

## 6. Traceability Identification and Planning

Traceability is an integrated system in which several stakeholders play a specific role. Furthermore, to implement a traceability system in the rice supply chain, it is essential to build internal traceability and chain traceability [26]. Hu et al. [12] explains that every stakeholder must be able to record all internal activities that enable users to trace back raw materials and trace forward the products. Data recorded on each stakeholder is then communicated to another stakeholder (chain traceability). The implementation of the traceability system can facilitate all stakeholders in diagnosing problems that occur during the production process and taking appropriate actions to rectify these problems.

An important task in developing a traceability system is identifying internal traceability information at each chain node. Therefore, the complexity in the development of traceability systems includes various stages such as planning, design and analysis, because it involves several stakeholders in the supply chain. According to [11,12] modeling using IDEF-0 can be used to develop internal traceability. Thakur et al. [6] explain that IDEF-0 is a modeling technique used to help identify, analyze and build business processes or software analysis. Furthermore, IDEF-0 is used to define request and design specifications for functions of a system. In this model, there are inputs, outputs, controls and mechanisms in a function. The IDEF-0 model for developing internal traceability systems is presented in Figure 3.

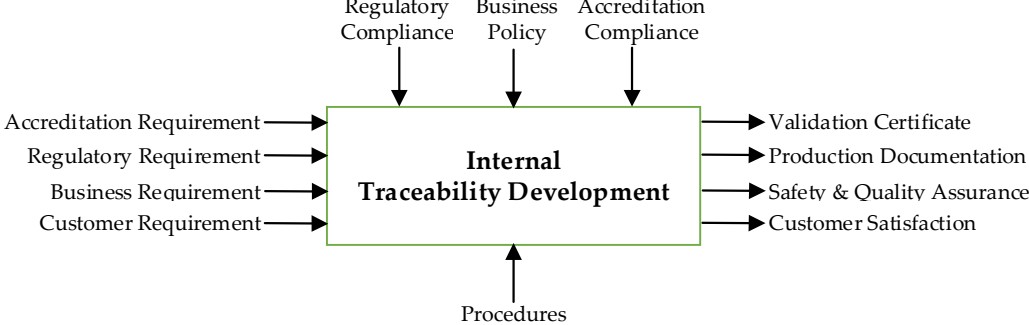

**Figure 3.** Integrated definition modeling (IDEF-0) for internal traceability development.

The main component in IDEF-0 is an analysis box that states the activities in the system and arrows that connect the activities. In the IDEF-0 diagram, horizontal lines inwardly and outwardly explain the input and output, while the vertical lines explain the controls and mechanisms for regulating

internal traceability. In designing the IDEF-0 framework, the main driving factor for developing internal traceability is regulation. Furthermore, industrial business needs and preferences are also documented in the model. According to Thakur et al. [6], the traceability system developed must follow the requirements set out in existing regulations. Compliance with rules becomes a control model along with guidelines from accreditation institutions and policies in business processes. The output of this model is in the form of documentation on the production process, quality assurance and food safety, validation certificates and providing customer satisfaction. Furthermore, the model is decomposed to show the stages in the system.

Decomposition is an activity that breaks down a system into its sub-components. The aim of decomposition is to define the whole process and identify relationships and dependencies between parts. Each level of abstraction provides more or fewer details about the entire system or a subset of the system. The IDEF-0 decomposition diagram shows all the stages involved in the internal traceability implementation process. This diagram shows the functional decomposition of the implementation of internal traceability in a top-down manner, which means detailing the operations from the system and sub-systems to specific events. This diagram is adjusted for all stakeholders in the supply chain and is built to obtain compliance with regulations, internal business policy, and certification from an accreditation institution. Figure 4 shows the IDEF-0 described, wherein at each stage, the diagram outlines the input, output, control, and mechanisms that occur.

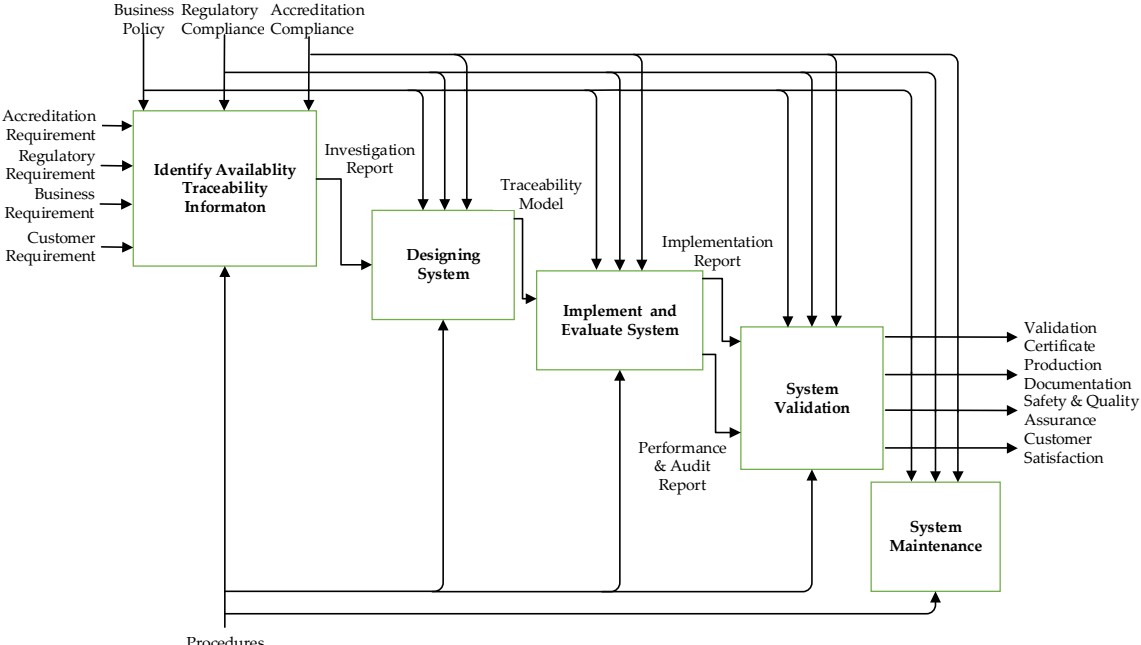

**Figure 4.** IDEF-0 implementation in internal traceability.

## 6.1. Identification Availability of Traceability Information

This stage includes the process of identifying the availability of information on internal traceability of all stakeholders in the supply chain. According to Thakur et al. and Kumar et al. [6,11], traceability information must clearly define the data captured from each stakeholder and the data shared with other stakeholders. In internal traceability, the data obtained from each stakeholder has a different purpose. Thus, it is necessary to identify the internal traceability of each stakeholder in order to meet the requirements set by each control from government regulations, accreditation institutions (such as the International Standardization Organization (ISO)) and internal policies. These controls have different characteristics for traceability, so they must be considered in planning and drafting a framework for implementing internal traceability. The output of this process is an investigative report that will be used to compile the traceability framework.

*6.2. System Design*

This stage tries to design a system that will be implemented. The rules of business processes for each stakeholder are determined, then the design of the Database Relation Management System (DBMS) is created using the Entity Relation Diagram (ERD) or Class Diagram (CD). This diagram is designed to show the relationship between entities or objects, operations or methods and their attributes in a system. The system framework, software architecture, interface design, operation methods, and system output are also made at this stage [10].

*6.3. Identification of Availability of Traceability Information*

The output of the second process is a system prototype that already has a database. Users can enter all information related to the production process from upstream to downstream. The database system will connect important information through a unique code or identification number. Furthermore, to optimize traceability, an evaluation is carried out to improve the efficiency and responsiveness of the system to the existence of a problem in food safety issues. After that, reports on system implementation, performance evaluation reports and system audits are produced. The control in this stage is the same as in the previous step.

*6.4. System Validation*

Validation is a step to determine whether a prototype is a representation of a real system using some evaluation metrics such as accuracy and Receiver Operating Characteristic (ROC) curves [27,28]. Validation is an approach to compare the output of a model with an actual system. Validation is conducted to ensure that the traceability system works well after the plan. In this stage, system validation is carried out based on regulatory controls including government regulations, accreditation institutions, and internal business policies as well as the same mechanism as before. Thakur [6] explains that with validation, compliance with controls can be achieved. Also, documentation of production, certification and quality management processes can be made. Consumer trust and satisfaction in food products will also increase.

*6.5. System Maintenance*

The system maintenance phase is carried out after the system implementation. System maintenance is an effort to maintain, repair, and update the existing system. Maintenance is carried out as long as the system functions to improve the efficiency and effectiveness of system performance because during its life span, there may be new problems and needs. According to Satzinger et al. [23], new requirements in traceability systems are related to adjustments to changes in regulations, business process requirements, customer needs, and other factors that can cause changes. This step is carried out continuously every time a change occurs in the product traceability process.

## 7. Design of Functional Structure of Traceability Systems

Internal traceability has an essential role in information exchange between actors in the supply chain. Every stakeholder must be able to document all relevant information related to the traceability unit. The development of a traceability information system requires a database that contains relevant information regarding traceability. Therefore, an approach is needed to be able to visualize the modeling of the data recorded. Modeling using an Entity Relationship Diagram (ERD) is the first step in designing a database. ERDs can be used to explain the relationship between data that is recorded and stored in a database based on fundamental data objects [29]. According to Ramadhan et al. [30], ERD is a representation of data requirements in a system during the system design process. This model describes the primary key (id) and data attributes contained in the system. Furthermore, ERD is a top-down approach to system design that starts with identifying the required data called entities and relationships between entities described in the form of models. Figure 5

describes an example of an ERD model that was built to facilitate an internal traceability system at a distributor.

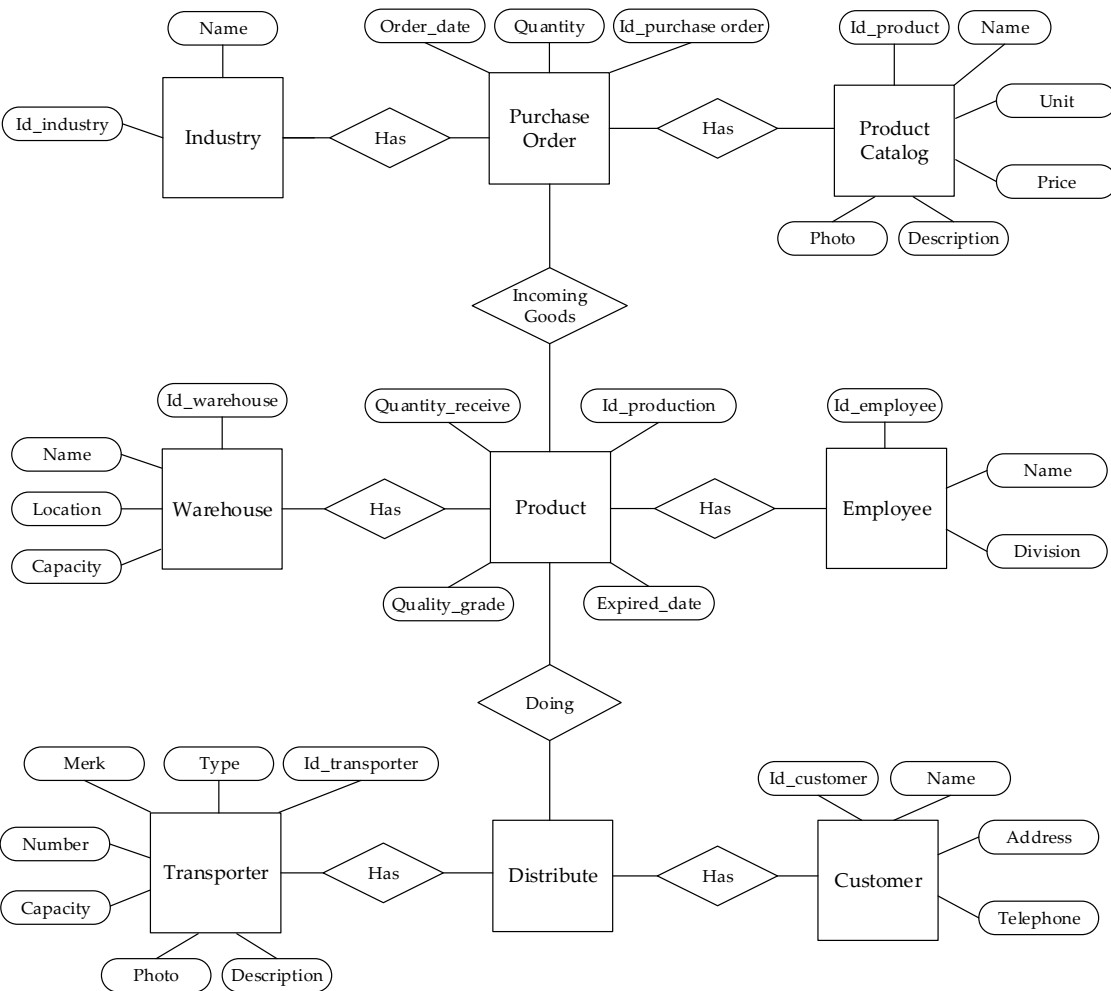

**Figure 5.** Entity relationship diagram (ERD) of internal traceability at the distributor.

## 8. Inter-Actor Traceability

Developing a traceability system requires integration and coordination between all stakeholders in the supply chain. Information exchange is the basis for integration and coordination among stakeholders in the supply chain and will realize a transparency process to manage risks that affect the quality and safety of the product. Information exchange in the supply chain enables all stakeholders to make appropriate decisions if indications are found that food products do not meet security requirements. Furthermore, effective information exchange among stakeholders can reduce the distortion and dissymmetry of information to have an impact on improving supply chain performance [7,31,32]. According to Golan et al. [33], transparency through the traceability system enables customers to choose products rationally from the various products available in the market. Information exchange in the supply chain can be used to monitor the production process and maintain partnerships between stakeholders [31,34,35]. In the traceability system, the most crucial thing in the supply chain is to record all relevant activities and to forward relevant information to the next stakeholder. Figure 6 shows the information that must be recorded and shared to the next stakeholder. Furthermore, based on Figure 6, not all information captured will be shared. The superscript shows the forwarded information to the next stakeholder, such as a lot number, variety, product ID and the others, to get detailed information throughout the traceability process.

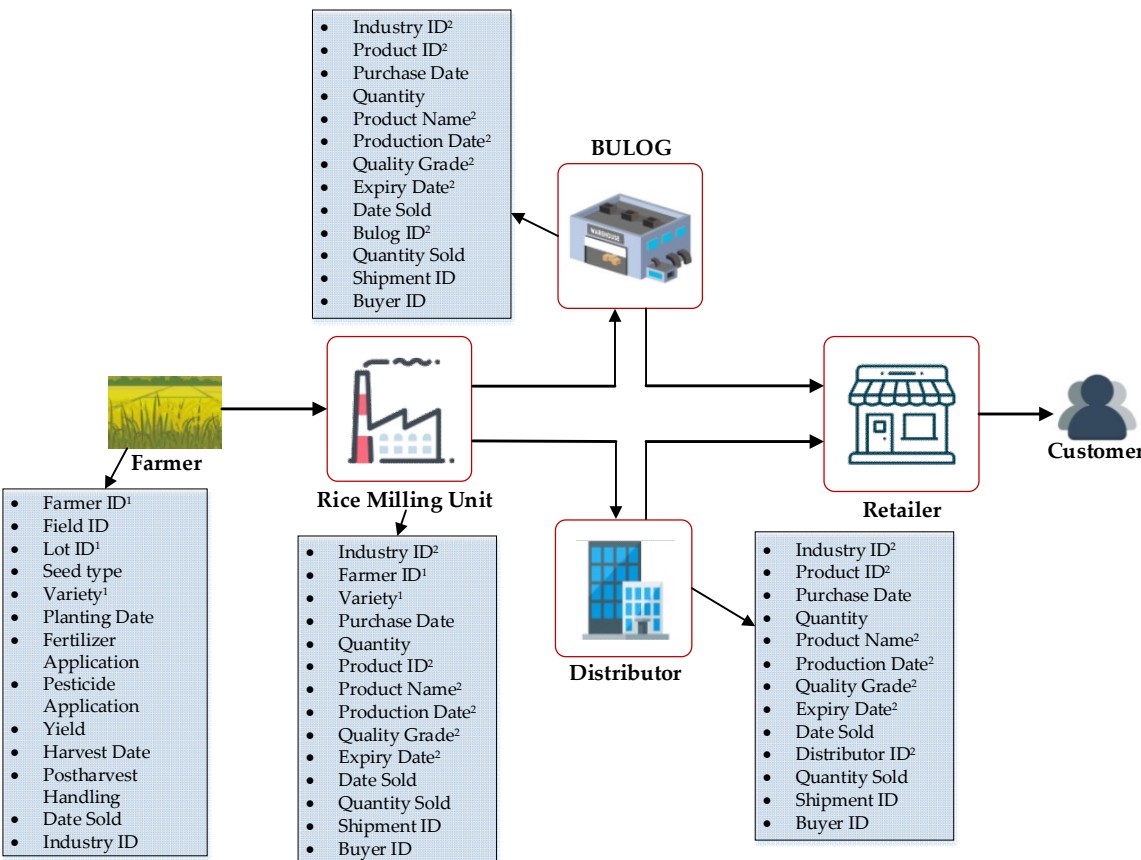

**Figure 6.** Forwarded information between stakeholders.

The information exchange model in the rice supply chain is developed to explain what information should be stored and exchanged. One step in modeling information exchange is to describe the exchange of information between stakeholders in the supply chain. Sequence diagrams explain how an object interacts with other objects. This diagram describes the behavior of objects in the use-case by describing the lifetime of objects and messages sent and received between objects. Sequence diagrams are used to communicate business processes by looking at interactions between stakeholders [23]. Generally, a sequence diagram that describes the exchange of information between stakeholders in the rice supply chain is presented in Figure 7.

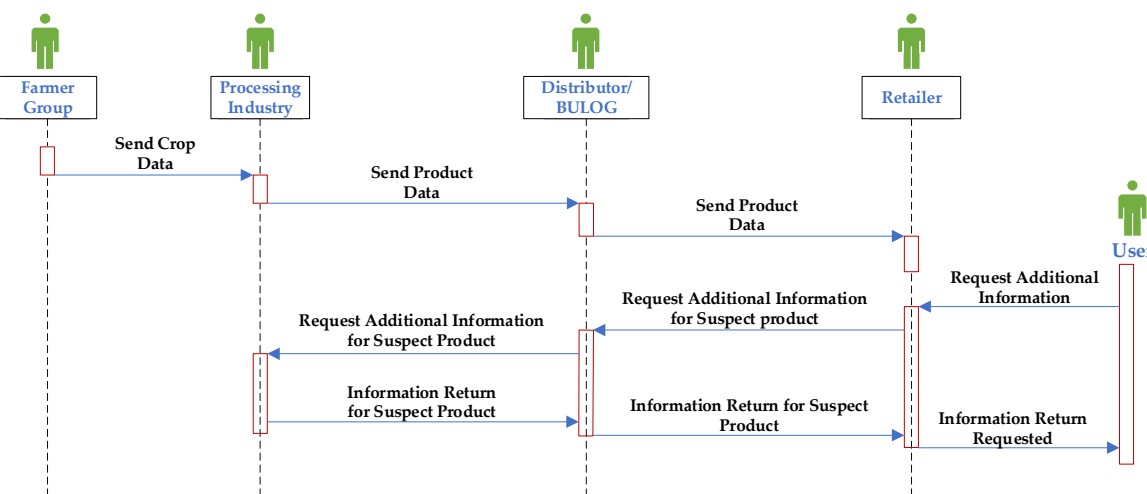

**Figure 7.** Sequence diagram for exchange of information in rice supply chains.

Thakur et al. [6] explain that the primary purpose of sequence diagrams is to define a series of events that produce the desired output. Figure 3 shows the sequence diagram when there are cases relating to food security, so the authorities can request additional information about products that are suspected of being dangerous. Therefore, all stakeholders in the supply chain must be able to provide the information needed in real time. The primary purpose of making sequence diagrams is to define activities that affect the outcome. This diagram shows the messages sent by each stakeholder using the traceability system [23]. Sequence diagrams convey information vertically and horizontally.

## 9. CBIS to Realize Traceability System

Generally, the traceability system requires all physical entities including the origin of the product, processing and packaging activity, and other processes in the supply chain [36]. Transparency in the rice supply chain enables a monitoring process for each stakeholder. Therefore, a system that records and documents the information is needed for all involved stakeholders. A CBIS (Computer Based Information System) is a model approach that can be used to provide the needed information. The application of CBIS allows all stakeholders to be connected with each other so that it will be easier to monitor each stakeholder. Furthermore, CBIS implementation enables traceability systems to be further developed by implementing ontology and also data mining by processing databases using statistical techniques or artificial intelligence to explore valuable business information such as product distribution, routing and scheduling [37,38], supplier selection [39], customer relationship management [40], and others. Figure 8 shows the CBIS framework in the traceability system built by adopting and modifying from Seminar [41].

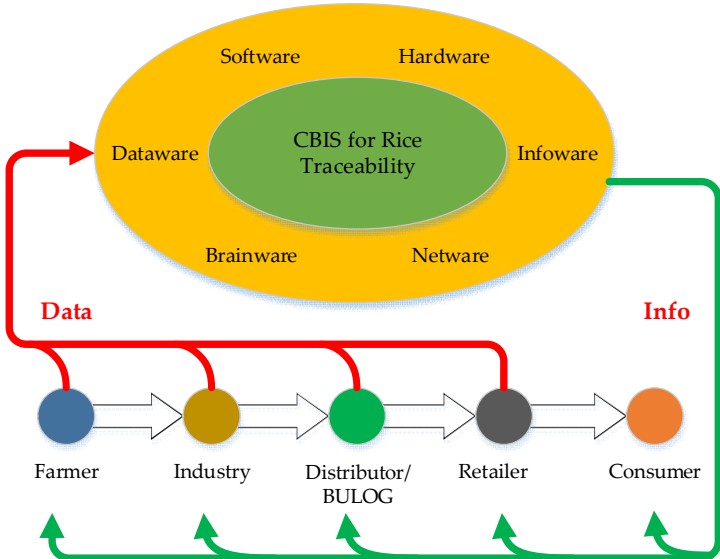

**Figure 8.** Computer-based information system (CBIS) for traceability system. Modified from Seminar [41].

To realize the transparency process in the rice supply chain, the application of a CBIS requires supporting resources consisting of: (1) hardware, which includes data recording devices, sensing devices, data scanning, communication devices, data storage, computers, and information display devices, (2) software, which includes database management systems (DBMS), data reporting applications, and geographic information system (GIS) applications, (3) dataware, which is all the data needed starting from location, actor data, activity data, industrial manufacturing data and quality data, (4) netware in CBIS, which includes access and network control, communication media used to integrate all stakeholders, network cluster technology (intranet, extranet and internet), (5) infoware, output in the form of information on a traceability system in digital or print out form that

can be used by stakeholders to help decision making and to obtain product information for customers, and (6) brainware, which consists of all stakeholders involved in the rice supply chain.

## 10. Conclusions

In order to realize traceability along rice supply chain, stakeholder must focus to internal and chain traceability. Identifying usage requirement is the first step in implementing traceability. Each stakeholder in rice supply chain must determine their traceability plan based on several parameters that infulence like regulatory requirement, business requirement, customer requirement, and accreditation requirement. In the process of designing internal traceability, modeling the captured data can be conducted by creating an Entity Relationship Diagram. All activities in the supply chain must be recorded and stored in the database management system, then several relevant data such as lot numbers must be passed on the next stakeholder. Additional information can be requested by a stakeholder in case unsafe products are found. Implementation of CBIS can be used in the development of a traceability system because it provides the ability to simplify the documentation process, integrate and facilitate information exchange between stakeholders. Furthermore, The future work is to start developing the rice supply chain traceability prototype ready for testing, validation, and deployment to finally produce the workable traceability system.

**Author Contributions:** This initial idea of this research was by K.B.S.; P.B.P. conducted an investigation to find out the field conditions. P.B.P. carried out system analysis and design; P.B.P. wrote the paper; K.B.S., (S.) Sutrisno, (S.) Sugiyanta reviewed and improved the paper.

**Acknowledgments:** This research was supported by the Ministry of Research, Technology and Higher Education of the Republic of Indonesia with contract number 1484/IT3.11/PN/2018 through the accelerated master's program leading to doctorate research grants (PMDSU). The authors also thank the reviewers who have provided constructive suggestions for the improvement of this paper.

**Conflicts of Interest:** The authors declare no conflict of interest.

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
