# Peer review of "Design Framework of a Traceability System for the Rice Agroindustry Supply Chain in West Java"

_information, doi:10.3390/info10060218_

Round 1

Reviewer 1 Report

The paper presents a framework for addressing the traceability system for rice within agroindustry supply chain in west java. following are my comments pertaining to the paper,

1. For readers to quickly catch the contribution in this work, it would be better to highlight major difficulties and challenges, and your original achievements to overcome them, in a clearer way in abstract and introduction.

2. The practical motivation of considering the rice transportation within agroindustry supply chain should be clearly and concretely presented.

3. Authors are advised to summarise the research gaps/issues by citing the relevant papers and it would be better if you could add contributions and motivation associated with the research work by linking them with the research gaps.

4. Author should provide certain managerial insights for justifying the proposed framework. Moreover,it will also help in understanding the importance of the framework in obtaining strategic decision making for an organization.

5. Please also discuss how sustainability can be address while developing this framework primarily addressing the traceability aspect. Some of the following papers from reputed journal need to be cited in your.

Hybridizing Basic Variable Neighborhood Search With Particle Swarm Optimization for Solving Sustainable Ship Routing and Bunker Management Problem, IEEE Transactions on Intelligent Transportation Systems

Multiobjective Approach for Sustainable Ship Routing and Scheduling With Draft Restrictions, IEEE Transactions on Engineering Management

A hybrid dynamic berth allocation planning problem with fuel costs considerations for container terminal port using chemical reaction optimization approach, Annals of Operations Research

6. Authors can discuss about how data mining and ontology techniques can be incorporated in their work for further extension in future. Following work of ontology need to be be referred in the paper. 

A decision support system based on ontology and data mining to improve design using warranty data, Computers and Industrial Engineering

Based on the above comments, I suggest major revision for the paper.

Author Response

Point 1: For readers to quickly catch the contribution in this work, it would be better to highlight major difficulties and challenges, and your original achievements to overcome them, in a clearer way in abstract and introduction.

Response 1:

Thanks for this suggestion.

We have fixed it according to your directives.

Abstract :The rice supply chain in Indonesia vary from one region to others and are difficult to trace movement along the chain from land to customers” (Have been described in our manuscript (line 12-13)

Introduction :In its implementation, traceability systems in several countries including Indonesia have not been carried out thoroughly from upstream to downstream and are still become the authority of each stakeholder, so that cases of quality manipulation, production errors, and other cases will be difficult to detect”. (Have been described in our manuscript (line 45-48)

Point 2: The practical motivation of considering the rice transportation within agroindustry supply chain should be clearly and concretely presented.

Response 2:

Thanks for this suggestion.

We have explain the motivation to design traceability framework in another way. We have fixed it on the introduction.

·In the first paragraph, we explain the general conditions of rice supply chain. (The blue highlights)

· In the second paragraph,  we explain the urgency of of the implementation of traceability system and the conditions for implementing traceability system in Indonesia. (The blue highlights)

·In the third paragraph, we explain the biggest challenge to design traceability system. (The blue highlights)

Rice is widely planted and is the staple food of around 3.5 billion people worldwide, and Indonesia is one of the countries with the highest consumption rates in the world which reached 38.41 million tons in 2012 [1,2]. In Indonesia, rice accounts for 45% of the total nutritional intake needed or about 80% of the primary carbohydrate sources in the consumption patterns of the Indonesian people [3]. Therefore, aspects of quality and food safety are essential things that must be considered. However, in reality, the rice production chain in Indonesia still faces several obstacles, and one of them is still the quality manipulation carried out by spesific stakeholders. Several previous studies conducted by Suismono and Ramli et al. [4,5] found rice that was sprayed with aromatic compounds and bleach with uncontrolled concentration where it could harm the health of consumers. Also, cases of counterfeiting of rice occur where traders or rice mills mix rice between varieties and between qualities.

Another important issue in the rice supply chain is the unavailability data that documenting all activities well starting from the production, processing and distribution. One of the suggested concepts to overcome these problems is recording all information about a food product through traceability system. Based on European Union regulations (178/2002), traceability is defined as the ability that can track and follow any food, feed, food producing animal, and other substances that will be used for consumption in all stages of production, processing and distribution [6-8]. In its implementation, traceability systems in several countries including Indonesia have not been carried out thoroughly from upstream to downstream and are still become the authority of each stakeholder, so that cases of quality manipulation, production errors, and other cases will be difficult to detect [9,10]. Furthermore, this problems will be detrimental to business actors (stakeholder) because consumers are hesitant to buy food products because there is no detailed information explicity needed regarding the quality and safety of these products [10]. It is crucial to develop Information Technology (IT)-based traceability system to overcome this problem, in the rice supply chain.

In the context of information exchange, IT-based traceability systems would support to communicate important information and integrate rice supply chains that are globally dispersed [11]. Nevertheless, the absence of a standard format for recording the information is a biggest challenge to realizing information exchange and integration between stakeholders along the supply chain. The design framework provides guidelines to stakeholders in the supply chain to streamline their operation process with each other to implement and maintain traceability [11]. This study was aimed to develop a design framework to implement internal and external traceability in the rice supply chain. Firstly the usage requirements of the system was define using use-case diagrams and IDEF-0 to show its implementation at each stakeholder. Furthermore, modeling the data requirements is described as the first step to create the database and sequence diagram was used to model the information exchange between stakeholders. Finally, the CBIS (Computer Based Information System) concept was proposed to support traceability along the rice supply chain”. (Have been described in our manuscript (line 29-64)

Point 3: Authors are advised to summarise the research gaps/issues by citing the relevant papers and it would be better if you could add contributions and motivation associated with the research work by linking them with the research gaps.

Response 3:

Thanks for this suggestion

We've followed your direction and add literature review to explain relevant paper and clarify the contribution of my study. (The blue highlights)

ISO 22005 explains that food security is the responsibility of all stakeholders involved in the production process. Therefore, all stakeholders in the supply chain must have the ability to identify who is the supplier (one step backward) and to whom the product is distributed (one step forward). The development of an Information Technology-based traceability system can be used because it has the ability to follow the historical route of food from upstream to downstream. This system is capable of capturing, storing, and transmitting information about the origin of raw materials, processing, and also all activities carried out by stakeholders in the supply chain so that it can ensure all production practices are carried out under the established standard operating procedures. The use of IT in traceability systems has several advantages, namely (a) integrating data and information from various stakeholders (multi-users), (b) increasing accuracy in data input, (c) providing the ability to communicate and exchange information between stakeholders, and (d ) simplify and speed up control and monitoring processes. Therefore, the use of IT can realize transparency in the supply chain because it is able to systematically manage information related to the products.

Nowadays information exchange become the main requirement to actualize effective and sustainable supply chain. However, Lam and Postle identified that the standard or framework that describes the process of information exchange on supply chain traceability systems is limited. In the food supply chain, several past studies on the development of traceability frameworks have been carried out by several researchers. Zhang et al. developed a traceability system design framework in the tilapia supply chain in China. In this study, stakeholders who play a role in the business process were identified and the supply chain structure was modeled. Furthermore, the information flow and the data that must be recorded by each stakeholder in the supply chain is described and then modeled with the UML (Unified Modeling Language) class diagram. Hu et al. designed a traceability system framework for vegetable supply chains in China. This study was able to identify the structure of vegetable supply chain and the relationship between stakeholders and traceability systems was modeled using use-case diagrams. A critical point analysis of valuable information was also carried out and then modeled using UML static diagrams. The results also explained the design and architecture of traceability systems in the vegetable supply chain. A system evaluation also carried out by Hu et al. for system improvement. From the perspective of the development of a traceability system, it is crucial to overcome traceability from data and information management, which relies heavily on the framework that has been prepared. However, not all traceability system frameworks have been available, primarily in the rice supply chain. (Have been described in our manuscript (line 66-97)

Point 4: Author should provide certain managerial insights for justifying the proposed framework. Moreover,it will also help in understanding the importance of the framework in obtaining strategic decision making for an organization.

Response 4:

Thanks for this suggestion.

We have explain on introduction.

The design framework provides guidelines to stakeholders in the supply chain to streamline their operation process with each other to implement and maintain traceability”. (Have been described in our manuscript (line 56-58).

Point 5: Please also discuss how sustainability can be address while developing this framework primarily addressing the traceability aspect. Some of the following papers from reputed journal need to be cited in your.

Response 5:

Thanks for this suggestion

We have explain on CBIS to realize traceability system.

Furthermore, CBIS implementation enables traceability systems to be further developed by implementing data mining by processing databases using statistical techniques or artificial intelligence to explore valuable business information such as product distribution routing and scheduling [37,38], supplier selection [39], customer relationship management [40], and others”. (Have been described in our manuscript (line 329-333).

Point 6: Authors can discuss about how data mining and ontology techniques can be incorporated in their work for further extension in future. Following work of ontology need to be be referred in the paper.

Response 6:

Thanks for this suggestion

We have explain on CBIS to realize traceability system.

Furthermore, CBIS implementation enables traceability systems to be further developed by implementing data mining by processing databases using statistical techniques or artificial intelligence to explore valuable business information such as product distribution routing and scheduling [37,38], supplier selection [39], customer relationship management [40], and others”. (Have been described in our manuscript (line 329-333).

Reviewer 2 Report

I would suggest the author to apply these chapters to organize this paper. Chapter 1 Introduction; Chapter 2 Literature Review; Chapter 3 Method; Chapter 4 Results; Chapter 5 Implications ; Chapter 6 Conclusions; and References. In present version, I think this might confuse the readers.

The three main criteria for this manuscript are: (a) quality and content of the research/review; (b) Quality, brevity and clarity of presentation; (c) Significance, relevance and timeliness of the topic. In addition, this title is (i) coverage of the literature/significant developments in the field, or clarity of discussion within an emerging topic; (ii) originality, new perspectives or insights; (iii) international interest; and (iv) relevance for governance, policy or practical perspectives relevant to the focus of this manuscript. However, this study is lacking most of important criteria. Hence, I think the author needs to consider these criteria before your submission.

Please make sure that a competent editor checks the English. Use of the first person (“I”, “we”, etc.) and third person ("he", "she" etc)must be avoided.

To be legible, the whole text must be completely edited with the help of a native English editor to polish your writing to prevent redundancies, grammatical errors and punctuation problems.

The abstract should state briefly the purpose of the research, the principal results and major conclusions. An abstract is often presented separately from the article, so it must be able to stand alone.

The major defect of this study is the debate or Argument is not clear stated in the introduction session. Hence, the contribution is weak in this manuscript. I would suggest the author to enhance your theoretical discussion and arrives your debate or argument.

The literature review is necessary for you to clarify the “contribution” of your study. In current form, there is none literatures to support your study. The author failed to present the study debates and failed to discuss the debates. In general, the author should present the specific debate for your study.

Mathematical formulation is logically and clearly presented. In addition, the case and associated data analysis are illustrative to demonstrate the usefulness of the proposed method.

I would like to request the author to emphasis on the contributions on practically and academically in implication section.

Please make sure your conclusions' section underscore the scientific value added of your paper, and/or the applicability of your findings/results, as indicated previously. Please revise your conclusion part into more details.  Basically, you should enhance your contributions, limitations, underscore the scientific value added of your paper, and/or the applicability of your findings/results and future study in this session.

Author Response

Point 1: I would suggest the author to apply these chapters to organize this paper. Chapter 1 Introduction; Chapter 2 Literature Review; Chapter 3 Method; Chapter 4 Results; Chapter 5 Implications ; Chapter 6 Conclusions; and References. In present version, I think this might confuse the readers.

Response 1:

Thank you, we have fixed it and added the literature chapter (have been described in our manuscript (line 65) and methodology (have been described in our manuscript (line 98).

Point 2: The three main criteria for this manuscript are: (a) quality and content of the research/review; (b) Quality, brevity and clarity of presentation; (c) Significance, relevance and timeliness of the topic. In addition, this title is (i) coverage of the literature/significant developments in the field, or clarity of discussion within an emerging topic; (ii) originality, new perspectives or insights; (iii) international interest; and (iv) relevance for governance, policy or practical perspectives relevant to the focus of this manuscript. However, this study is lacking most of important criteria. Hence, I think the author needs to consider these criteria before your submission.

Response 2:

Thanks for this suggestion.

We have modified it to explain three main criteria on introduction and literature review.

Point 3: Please make sure that a competent editor checks the English. Use of the first person (“I”, “we”, etc.) and third person ("he", "she" etc)must be avoided.

Response 3:

Thank you, we have fixed it according to your directives.

Point 4: To be legible, the whole text must be completely edited with the help of a native English editor to polish your writing to prevent redundancies, grammatical errors and punctuation problems.

Response 4:

Thank you, we have fixed it according to your directives, and we also provide the English editing certificate.

Point 5: The abstract should state briefly the purpose of the research, the principal results and major conclusions. An abstract is often presented separately from the article, so it must be able to stand alone.

Response 5:

Thanks for this suggestion

We have modified the abstract for this article correspond to your direction :

Rice is a vital food commodity in Indonesia due to its role as a staple food for most of the Indonesian people. The rice supply chain in Indonesia vary from one region to others and are difficult to trace movement along the chain from land to customers. This introduces untransparency and uncertainty on the quantity and quality of rice at every node along the supply chain. The crucial issues on food safety and security, as well as consumers concern and curiousity in buying and consuming foods, increases the need a traceability system for rice value chain which can be easily and widely accessed. This paper describes the design framework of an IT-based traceability system for the rice supply chain on web platforms. The system approach has been followed, where the system requirements are identified based on supply chain characteristics, and then the logical framework for implementing internal and external traceability is modeled using IDEF-0 (Integrated Definition Modeling). This paper further presents an explanation of ERD (Entity Relationship Diagram) as an initial step to modeling data requirements and a model of information exchange between stakeholders explains the data that must be recorded and forwarded to the next stakeholder. Finally, we propose the CBIS (Computer Based Information System) concept to develop a traceability system in the rice supply chain. (Have been described in our manuscript (line 11-25)

Point 6: The major defect of this study is the debate or Argument is not clear stated in the introduction session. Hence, the contribution is weak in this manuscript. I would suggest the author to enhance your theoretical discussion and arrives your debate or argument.

Response 6:

Thanks, we’ve followed your direction to totally fix the introduction and explain the debate or argument for this session as follows.

·      In the first paragraph, we explain the general conditions of rice supply chain. (The blue highlights)

·      In the second paragraph,  we explain the urgency of of the implementation of traceability system and the conditions for implementing traceability system in Indonesia. (The blue highlights)

·      In the third paragraph, we explain the biggest challenge to design traceability system. (The blue highlights)

Rice is widely planted and is the staple food of around 3.5 billion people worldwide, and Indonesia is one of the countries with the highest consumption rates in the world which reached 38.41 million tons in 2012 [1,2]. In Indonesia, rice accounts for 45% of the total nutritional intake needed or about 80% of the primary carbohydrate sources in the consumption patterns of the Indonesian people [3]. Therefore, aspects of quality and food safety are essential things that must be considered. However, in reality, the rice production chain in Indonesia still faces several obstacles, and one of them is still the quality manipulation carried out by spesific stakeholders. Several previous studies conducted by Suismono and Ramli et al. [4,5] found rice that was sprayed with aromatic compounds and bleach with uncontrolled concentration where it could harm the health of consumers. Also, cases of counterfeiting of rice occur where traders or rice mills mix rice between varieties and between qualities.

Another important issue in the rice supply chain is the unavailability data that documenting all activities well starting from the production, processing and distribution. One of the suggested concepts to overcome these problems is recording all information about a food product through traceability system. Based on European Union regulations (178/2002), traceability is defined as the ability that can track and follow any food, feed, food producing animal, and other substances that will be used for consumption in all stages of production, processing and distribution [6-8]. In its implementation, traceability systems in several countries including Indonesia have not been carried out thoroughly from upstream to downstream and are still become the authority of each stakeholder, so that cases of quality manipulation, production errors, and other cases will be difficult to detect [9,10]. Furthermore, this problems will be detrimental to business actors (stakeholder) because consumers are hesitant to buy food products because there is no detailed information explicity needed regarding the quality and safety of these products [10]. It is crucial to develop Information Technology (IT)-based traceability system to overcome this problem, in the rice supply chain.

In the context of information exchange, IT-based traceability systems would support to communicate important information and integrate rice supply chains that are globally dispersed [11]. Nevertheless, the absence of a standard format for recording the information is a biggest challenge to realizing information exchange and integration between stakeholders along the supply chain. The design framework provides guidelines to stakeholders in the supply chain to streamline their operation process with each other to implement and maintain traceability [11]. This study was aimed to develop a design framework to implement internal and external traceability in the rice supply chain. Firstly the usage requirements of the system was define using use-case diagrams and IDEF-0 to show its implementation at each stakeholder. Furthermore, modeling the data requirements is described as the first step to create the database and sequence diagram was used to model the information exchange between stakeholders. Finally, the CBIS (Computer Based Information System) concept was proposed to support traceability along the rice supply chain. (Have been described in our manuscript (line 29-64)

Point 7: The literature review is necessary for you to clarify the “contribution” of your study. In current form, there is none literatures to support your study. The author failed to present the study debates and failed to discuss the debates. In general, the author should present the specific debate for your study.

Response 7:

Thanks, we've followed your direction to add literature review to clarify the contribution of my study. (The blue highlights)

ISO 22005 explains that food security is the responsibility of all stakeholders involved in the production process. Therefore, all stakeholders in the supply chain must have the ability to identify who is the supplier (one step backward) and to whom the product is distributed (one step forward). The development of an Information Technology-based traceability system can be used because it has the ability to follow the historical route of food from upstream to downstream. This system is capable of capturing, storing, and transmitting information about the origin of raw materials, processing, and also all activities carried out by stakeholders in the supply chain so that it can ensure all production practices are carried out under the established standard operating procedures. The use of IT in traceability systems has several advantages, namely (a) integrating data and information from various stakeholders (multi-users), (b) increasing accuracy in data input, (c) providing the ability to communicate and exchange information between stakeholders, and (d ) simplify and speed up control and monitoring processes. Therefore, the use of IT can realize transparency in the supply chain because it is able to systematically manage information related to the products.

Nowadays information exchange become the main requirement to actualize effective and sustainable supply chain. However, Lam and Postle identified that the standard or framework that describes the process of information exchange on supply chain traceability systems is limited. In the food supply chain, several past studies on the development of traceability frameworks have been carried out by several researchers. Zhang et al. developed a traceability system design framework in the tilapia supply chain in China. In this study, stakeholders who play a role in the business process were identified and the supply chain structure was modeled. Furthermore, the information flow and the data that must be recorded by each stakeholder in the supply chain is described and then modeled with the UML (Unified Modeling Language) class diagram. Hu et al. designed a traceability system framework for vegetable supply chains in China. This study was able to identify the structure of vegetable supply chain and the relationship between stakeholders and traceability systems was modeled using use-case diagrams. A critical point analysis of valuable information was also carried out and then modeled using UML static diagrams. The results also explained the design and architecture of traceability systems in the vegetable supply chain. A system evaluation also carried out by Hu et al. for system improvement. From the perspective of the development of a traceability system, it is crucial to overcome traceability from data and information management, which relies heavily on the framework that has been prepared. However, not all traceability system frameworks have been available, primarily in the rice supply chain. (Have been described in our manuscript (line 66-97)

Point 8: Mathematical formulation is logically and clearly presented. In addition, the case and associated data analysis are illustrative to demonstrate the usefulness of the proposed method.

Response 8:

Generally, the design framework of traceability system we must (1) analyzing business processes in the rice supply chain and (2) modeling the system requirement using descriptive and qualitative analysis.

Point 9: I would like to request the author to emphasis on the contributions on practically and academically in implication section.

Response 9:

With all the limitations form the author, the chapter on the implication has not been compiled.

Point 10: Please make sure your conclusions' section underscore the scientific value added of your paper, and/or the applicability of your findings/results, as indicated previously. Please revise your conclusion part into more details.  Basically, you should enhance your contributions, limitations, underscore the scientific value added of your paper, and/or the applicability of your findings/results and future study in this session.

Response 10:

Thanks for this suggestion

We have modified the conclusion for this article correspond to your direction :

“In order to realize traceability along rice supply chain, stakeholder must focus to internal and chain traceability. Identifying usage requirement is the first step in implementing traceability. Each stakeholder in rice supply chain must determine their traceability plan based on several parameters that infulence like regulatory requirement, business requirement, customer requirement, and accreditation requirement. In the process of designing internal traceability, modeling the captured data can be conducted by creating an Entity Relationship Diagram. All activities in the supply chain must be recorded and stored in the database management system, then several relevant data such as lot numbers must be passed on the next stakeholder. Additional information can be requested by a stakeholder in case unsafe products are found. Implementation of CBIS can be used in the development of a traceability system because it provides the ability to simplify the documentation process, integrate and facilitate information exchange between stakeholders. Furthermore, The future work is to start developing the rice supply chain traceability prototype ready for testing, validation, and deployment to finally produce the workable traceability system”. (Have been described in our manuscript (line 352-364)

Round 2

Reviewer 1 Report

Authors have adequately addressed the comments and now the paper can be accepted for publication in the journal.

Reviewer 2 Report

Authors have been revised their study in accordance with reviewer's comments, thus, I would like to see this study that published in the Information.